# A Systematic Review and Meta-Analysis of the Differences in Mean Propulsive Velocity between Men and Women in Different Exercises

**DOI:** 10.3390/sports11060118

**Published:** 2023-06-13

**Authors:** Raúl Nieto-Acevedo, Blanca Romero-Moraleda, Francisco Javier Díaz-Lara, Alfonso de la Rubia, Jaime González-García, Daniel Mon-López

**Affiliations:** 1Departamento de Deportes, Facultad de Ciencias de la Actividad Física y del Deporte, Universidad Politécnica de Madrid, Calle Martín Fierro, 7, 28040 Madrid, Spain; nietoacevedoraul@gmail.com (R.N.-A.); alfonso.delarubia@upm.es (A.d.l.R.); daniel.mon@upm.es (D.M.-L.); 2Department of Physical Education, Sport and Human Movement, Autonomous University of Madrid, 28049 Madrid, Spain; blanca.romero@uam.es; 3Performance and Sport Rehabilitation Laboratory, Faculty of Sports Sciences, University of Castilla-La Mancha, Avda. Carlos III S/N, 45071 Toledo, Spain; javier.diazlara@uclm.es; 4Exercise and Sport Sciences, Faculty of Health Science, Universidad Francisco de Vitoria, 28223 Pozuelo, Spain

**Keywords:** load–velocity profile, mean propulsive velocity, sex differences, velocity-based training

## Abstract

The purpose of this paper was to conduct a systematic review and meta-analysis of studies examining the differences in the mean propulsive velocities between men and women in the different exercises studied (squat, bench press, inclined bench press and military press). Quality Assessment and Validity Tool for Correlational Studies was used to assess the methodological quality of the included studies. Six studies of good and excellent methodological quality were included. Our meta-analysis compared men and women at the three most significant loads of the force–velocity profile (30, 70 and 90% of 1RM). A total of six studies were included in the systematic review, with a total sample of 249 participants (136 men and 113 women). The results of the main meta-analysis indicated that the mean propulsive velocity is lower in women than men in 30% of 1RM (ES = 1.30 ± 0.30; CI: 0.99–1.60; *p* < 0.001) and 70% of 1RM (ES = 0.92 ± 0.29; CI: 0.63, 1.21; *p* < 0.001). In contrast, for the 90% of the 1RM (ES = 0.27 ± 0.27; CI: 0.00, 0.55), we did not find significant differences (*p* = 0.05). Our results support the notion that prescription of the training load through the same velocity could cause women to receive different stimuli than men.

## 1. Introduction

Evidence has shown that improvements in muscular strength are related with a higher performance in general sports skills such as jumping, sprinting, and change of direction tasks [1]. In order to increase muscular strength, it is necessary to manipulate training variables appropriately. Specifically, exercise intensity is one of the most critical variables in resistance training and is usually reported as a percentage of the individuals one repetition maximum (1RM) (e.g., 80% 1RM) [2]. Velocity-based training (VBT) can be used to accurately determine the intensity of training [3]. VBT uses the velocity of the bar to determine the relative load, and there are many studies that calculated the velocity of the bar for each percentage of 1RM in different exercises: prone bench pull [4], pull-up [5,6], leg press [7], hip thrust [8], and a number of variations of the squat and bench press exercise [9,10], finding differences between exercise in the velocity associated to each percentage of 1RM (e.g., 70% 1RM in the squat: 0.73 ± 0.05 m/s vs. 70% 1RM in the bench press: 0.58 ± 0.08) (Table 1). In this sense, differences in the range of motion between exercises could affect the rate of force development, activation and synchronization of motor units [11].

Additionally, one of the problems of VBT is that most previous studies only include men, and response to training can be different between men and women [3]. Although the underlying mechanisms are not fully understood, the proportional difference in fiber type between sexes can influence the contractile properties of skeletal muscle in men and women [12]. For example, male skeletal muscle exhibits faster relaxation rates than female muscle [13], consistent with a greater proportional area of type I fibers in females. Additionally, contrary to women, for men, testosterone levels do not show concentration fluctuations during the day [14]. Furthermore, evidence suggests that the response to the same ‘dosage’ of exercise could be different between males and females due to skeletal muscle’s metabolic and contractile properties [15]. Taking into account the previous literature and the growing interest in VBT, the general equations previously published for the main exercises may not be appropriate for women. Then, prescribing the training load through the same velocity could lead to women receiving different stimulus than men. In fact, some studies have shown that men and women respond differently to resistance training. For example, in a systematic review with methanalysis [16], the authors found a moderate effect size favoring females in the upper-body strength analysis, which makes it possible for untrained females to show a higher capacity to increase upper-body strength compared to males. Moreover, the results of study [17] suggest that men and women experience similar strength and power increase but achieve this using different levels of velocity loss (VL) (women 40% VL and men 20% VL). Therefore, women may benefit from greater volume during resistance training compared to men [18]. In addition, women seem to report lower levels of acute neuromuscular fatigue than men when performing the same loading protocol [19,20]. Another consideration regarding response to training in males and females is that pre-training levels of muscle size and strength are generally greater in males, independent of training status [16]. In addition, it is well known that women experience significant hormonal alteration across the menstrual cycle, which may influence muscle hypertrophy and strength adaptations [21,22]. However, the results of study [23] suggest that the estimation of the bench press 1RM from the load–velocity relationship seems not to vary over the three different phases of the menstrual cycle.

Due to these data, knowledge of the differences between men and women in the exercises studied could aid in using sex-specific data in order to make training as individualized as possible. The absence of this information is a research gap; filling it would allow programing based on the intensity of the load more accurately using VBT for females. Therefore, we hypothesized that men have higher mean propulsive velocities (MPV) than women in the different exercises studied.

## 2. Materials and Methods

### 2.1. Search Strategy and Study Selection

This systematic review and meta-analysis was carried out in accordance with the recommendations described in the Preferred Reporting Items for Systematic Reviews and Meta-Analysis (PRISMA) statement [24]. A search was performed from the oldest record up to and including November 2022 using the following electronic databases: PubMed, SportDiscus, and Web of Sciences (WOS). The search strategy used combined the terms ‘load–velocity profile’ OR ‘load–velocity relationship’ OR ‘load–velocity profiling’ OR ‘power–velocity relationships’ OR ‘load–velocity profiles’ OR ‘Velocity-Based Methods’. The corresponding authors of potentially eligible articles have been contacted to obtain missing data or clarify the presented data. The search for published studies was independently performed by two authors (R.N.A and F.J.D.L) and disagreements were resolved through discussion. We previously registered the protocol of this systematic review on PROSPERO (CRD42020180911).

### 2.2. Inclusion Criteria

Our analysis was limited to studies published in English-language peer-reviewed journals that met the following criteria: (1) the study involved both gender participants (men and women), (2) the study included the use of the mean propulsive velocity method to estimate the load–velocity profile, and (3) the intensity in the exercises ranged from 30% 1RM to 100% 1RM.

### 2.3. Data Extraction

The following information was extracted from the included studies: (1) author and year of study; (2) sample; (3) age of participants; (4) training status; (5) exercise tested; (6) main findings. When needed, the Web Plot Digitizer software (V.3.11, Texas, USA) was used to extract data from figures. Standard error values reported in some studies were transformed to standard deviation.

### 2.4. Methodological Quality

All articles were screened for quality using the published Quality Assessment and Validity Tool for Correlational Studies adapted from previous systematic reviews [25,26,27] (Figure 1).

The instrument included 13 questions to scrutinize and score the research design, sample, measurement and statistical analysis of the studies. The questions were in the dichotomous answer format, and a total of 14 points could be assigned for the 13 criteria. Twelve items were scored as 0 (=NO) or 1 (=YES) and the item related to outcome measurement were scored out of two. Based on the scores assigned, the instrument classifies the articles into three quality categories: low (0–4), medium (5–9) and high (10–14). Two authors (the first and second author) performed the appraisal of methodological quality independently. Any differences in the assessment between the authors were resolved through a third author who acted as a referee.

### 2.5. Statistical Analyses

All meta-analyses were performed using the random-effects model. The statistical significance threshold was set at *p* < 0.05 for all statistical analyses. The data analyses were performed using Review Manager (5.3, London, UK) [28]. The meta-analyses comparing the mean propulsive velocity outcomes between males and females were carried out using standardized mean differences (SMDs) and their respective 95% confidence intervals (95% CI). For each outcome, the SMD was calculated using mean and standard deviation values from a sample size from each study, finding the mean values in different exercises from men and women. The mean propulsive velocity was analyzed on the following exercises: (1) bench press; (2) press horizontal; (3) inclined bench press; (4) military press and (5) squat. The present meta-analysis includes a total of three studies [29,30,31] with six exercises where the loads of 30%, 70% and 90% 1RM were examined according to the optimal load to produce power [32,33]. The magnitude of the difference in MPV between males and females was interpreted by using the SMD scale: <0.2 (trivial); 0.2–0.6 (small); 0.6–1.2 (moderate); 1.2–2.0 (large); 2.0–4.0 (very large); and >4.0 (extremely large) (Hopkins et al. 2009). Heterogeneity was assessed using the I2 statistic and interpreted as follows: 0–40% (might not be important); 30–60% (may represent moderate heterogeneity); 50–90% (may represent substantial heterogeneity); and 75–100% (considerable heterogeneity) [34].

## 3. Results

### 3.1. Search Results

The search through the three databases produced a total of 39 search results. Of that number, seven full-text papers were read. Of the seven read studies, six satisfied the inclusion criteria [30,31,35,36]. Six studies were published in peer-reviewed journals. The flow diagram of the search is presented in Figure 2.

### 3.2. Study Characteristics

The pooled number of participants in the six studies is 249 (men *n* = 136 [age: 21.2 ± 2.6], women *n* = 113 [age: 21.53 ± 2.55]). In two studies, the sample consisted of rugby players or judoka. All included studies used a linear velocity transducer (T-Force System; Ergotech, Murcia, Spain) for the testing sessions. All studies except one [29] reported at least 2 years of resistance training experience with exercise testing. The summary of all included studies is presented in Table 1.

**Table 1 sports-11-00118-t001:** Summary of the included studies.

Reference	N	Age	Training Status	Exercise	Major Finding
Pareja-Blanco et al. [29]	M: 25	25.8 ± 3.3	1 year of RT	Squat and Bench Press	Significant differences between the sexes (30 to 75% 1RM) on Squat and Bench Press.
	W: 25	26.1 ± 4.0			
Torrejón et al. [30]	M: 14	23.8 ± 2.5	Men presented higher experience with the bench press exercise than women	Bench Press	The MV associated with the light loads (≈30% 1RM) was higher for men, women presented higher MV values for heavy loads (≈100% 1RM).
	W: 14	21.5 ± 1.4			
Balsalobre et al. [35]	M: 26	21.2 ± 3.8	At least 2 years of experience with the exercise	Seated military press	Moderate differences in the load–velocity profile between men and women. Namely, men showed higher MPV values at different % 1RM than women, with the exception of MPV at 100%1-RM in which the differences between males and females were trivial.
	W: 13	22.3 ± 3.3			
García-Ramos et al. [31]	M: 12	19.9 ± 1.2	At least 2 years of resistance training experience (2–5 sessions/week)	Horizontal bench press (HBP); Inclined bench press; (IBP) Seated military press (SMP)	Men showed higher velocities than women for the same percentaje of 1RM during the three exercises (20 to 75% 1RM). The load–velocity profile is exercise-specific.
	W: 12	20.4 ± 1.3			
Iglesias-soler et al. [37]	M: 13	22 ± 3	4 rugby players and 4 judokas	Squat and Bench Press	The main finding of this study was that multilevel mixed regression models detected significant inter-individual variance in the slopes and intercepts of the LV relationship. Additionally, for BP, including sex as a fixed factor improved the goodness of fit, but this was not the case when the interaction between sex and percentaje of 1RM was added.
	W: 8	24 ± 2	5 rugby players and 8 judokas		
Alonso-Aubin Diego A. et al. (2019) [36]	M: 46	14.5 ± 1.3	Rugby players	Squat and Bench Press	Significant sex-related differences were found on the squat exercise for maximum, mean, and time to maximum velocities as well as time to maximum power. On the bench press exercise, significant sex-related differences were found for power and time to maximum velocity.
	W: 41	14.9 ± 2.8			

M: Men; W: Women; RT: Resistance Training; MV: Mean Velocity; MPV: Mean Propulsive Velocity; LV: Load Velocity. Data are presented as mean ± SD.

### 3.3. Summary of Quality

The average methodological quality score was 9 (range 9 to 11). Based on these scores, two studies included were classified as high quality while four studies were rated medium quality. Most of the studies utilized a non-experimental, cross-sectional design. All studies were prospective in design as data were collected prospectively. All studies satisfied the items 1, 6, 7, 8, 9, 11, 12. Two studies [29,30] did not use a probability sample (Item 2). In addition, they did not justify the sample size [35,36,37] (Item 3). We considered that three studies [29,30,35] did not safisfy Item 4 because they obtained samples for more than one site. Most of the included studies [29,30,35,36] did not report interal consistency (Item 10). Three studies [35,36,37] did not mention anonymity protection; therefore, we considerer it unclear whether Item 5 was satisfied. Only one study [31] reported outlier values (Item 13). The results of the quality assessment of the included studies can be found in Table 2.

### 3.4. Meta-Analysis Results

Results showed that significantly higher MPV favored men with 30% 1RM (ES = 1.30 ± 0.30; CI: 0.99–1.60; *p* < 0.001; I = 1; Figure 3) and 70% 1RM (ES = 0.92 ± 0.29; CI: 0.63, 1.21; *p* < 0.001; Figure 4), but non-significant differences were revealed with 90% 1RM (ES = 0.27 ± 0.27; CI: 0.00, 0.55; *p* = 0.05; Figure 5). In addition, women showed lower MPV value than men for all the mean loads (30, 70 and 90% 1RM) (ES = 0.99 ± 0.16; CI: 0.63, 1.35; *p* < 0.001; I = 32; Figure 6).

Comparing men and women in terms of different loads (30, 70 and 90% 1RM), we determined that the differences decrease when the load is increased in squat (SMD: 1.27 vs. 1.13; 0.73 vs. 0.55; 0.46 vs. 0.42, respectively). A similar situation is observed in bench press (SMD: 1.25 vs. 1.08; 0.62 vs. 0.55; 0.30 vs. 0.29, respectively), inclined bench press (SMD: 1.30 vs. 1.14; 0.66 vs. 0.60; 0.34 vs. 0.33, respectively) and military press (SMD: 1.36 vs. 1.25; 0.69 vs. 0.65; 0.36 vs. 0.36, respectively) exercises (*p* < 0.05) (Figure 3, Figure 4 and Figure 5).

Comparison by muscle groups analyzing both sexes together showed higher MPV in squat (SMD: 1.62; *p* < 0.001) vs. bench press (SMD: 0.68; *p* < 0.001) exercises when the mean all loads were compared (Figure 6). These differences are lower in comparisons between bench press, inclined bench and press military press (SMD: 1.40 > 1.07 > 0.61, respectively; *p* < 0.001).

## 4. Discussion

The main aim of this meta-analysis was to compare mean propulsive velocities between men and women in the different exercises. We found that MPV is lower in women than men in 30 and 70% of 1RM with a large effect (SMD: 1.30, 0.92, respectively). Similarly, for the 1RM mean of all loads, men reported higher velocities than women (SMD: 0.99). On the contrary, our analysis did not find differences at 90% of 1RM (SMD: 0.27). Consequently, our results suggest that there are considerable sex differences in MPV associated with different percentages of 1RM in various resistance exercises. Apparently, these gender differences seem to tend to decrease as the load increases, as we observed at 30, 70 and 90% 1RM (SMD: 1.30 > 0.92 > 0.27, respectively). Another interesting result was the differences between exercises that showed higher MPV in squat vs. bench press (SMD: 1.62 vs. 0.68) exercises when comparing the mean loads of all. Although these differences are minor in comparisons between bench press, inclined bench and press military press (SMD: 1.40 > 1.07 > 0.61, respectively) exercises.

Our results confirm the differences between men and woman, especially in 30% and 70% 1RM. For this percentaje of 1RM, we found larger differences between sex in the bench press, inclined bench press, and military press exercises in all the studies included in our meta-analysis. These differences between exercises may be due to the fact that bench press involves larger muscle groups, which may accentuate the differences between men and women. In this line, some studies show that men perform more repetitions with the same relative load in the bench press [38] or that men lift 2.4 times more weight relative to their body than women [39]. Another possible reason could be that the angle of the bench produces a significantly higher activation of the anterior deltoid and decreases the muscle performance of the pectoralis major [40].

Another major result was the difference between upper limb and lower limb exercises. We found a higher differences in squat (SMD: 1.62) vs. bench press (SMD: 0.68). These results are in line with those of Nikoladis, P.T. et al. [41] who determined that arms had lower values of power and force compared to legs, and smaller differences concerning velocity.

This fact could be associated to the fact that the squat exercise involves more body mass and men usually have more total weight than women. In this line, Andrew et al. [42] found that men produced higher absolute peak power, average power, peak velocity, and average velocity across all loads used. Sex differences in peak power and average power seemed to be strongly related to body mass and 1RM. Similarly, Iglesias Soler et al. [37] support the fact that differences in power between men and women disappear when body mass is included in the formula. This could also be attributed in part to the fact that women have lower muscle mass, which allows greater muscle perfusion [43] and greater type I muscle fiber content [44]. Moreover, another possible reason for the differences between sexes could be the range of motion (ROM). In general, men are taller and have longer limbs than women [45]. Some studies [46,47,48] have shown that variations in the ROM of the concentric phase influences several biomechanical factors and can affect the development of force, rate of force development, and activation and synchronization of motor units. Martínez-Cava et al. [46] showed that the ROM influences MPV. The authors found that ROM affected the 1RM strength, load–velocity profiles and the contribution of the propulsive phase. This could partly explain the gender differences in MPV due to differences in limb lengths between men and women. Interestingly, the results of study [49] reveal that humerus length is significantly correlated to the average concentric velocity values at moderate loads for the upper-body exercises, whereas femur length is not related to the average concentric velocity values for the lower-body exercises. However, height is related to the average concentric velocity at various loads in squat, bench press, deadlift, and overhead press [49]. In addition, neuromuscular responses are one factor that could explain the differences in strength between sexes. In this sense, one study [50] compared the number of motor units in the biceps brachii and vastus medialis, finding differences between sexes.

On the contrary, for the 90% 1RM, we did not detect significant differences. These results are similar to the findings of Pareja-Blanco et al. [29] who did not observe significant differences by sex in heavy loads. In addition, Balsalobre et al. [35] reported that there were no differences between sexes with 100% 1RM in seated military press. In the same line, García Ramos et al. [31] did not report differences between men and women for 1RM velocities in military press exercises. Considering these results, it seems that the differences between sexes in MPV disappear with submaximal loads (~90% 1RM). These results are in line with those obtained by Soriano et al. [51], who found that sex does not affect the differences in the 1RM performance across weightlifting overhead pressing exercises. This could be explained by the Henneman Size Principle [52], which states that with maximal or submaximal loads the velocity is lower and, consequently, the participation of slow fibers is greater. This would eliminate the advantage of men over women of having more fast fibers [12].

Unfortunately, most studies on mean propulsive velocity have been performed on subjects of the male population. To the best of our knowledge, there are just three studies directly analyzing the differences between men and women in regard to velocity profiles using mean propulsive velocity [29,35,53]. The main limitation of our meta-analysis is that we only included three studies that analyze six exercises because there are few studies comparing mean propulsive velocity between men and women. In addition, the training status of the participants varied considerably, and a wide spectrum was included in the analysis. This is a limitation because some studies have shown tthar relative strength is positively related with average concentric velocity [49]. For example, the average concentric velocity during the 1RM squat in study [49] (0.26 ± 0.08 m/s^1^) was slightly lower than that of the studies performed using novice squatters [54], recreationally trained men [55,56] and college athletes [57], but slightly higher than that of the studies analyzing powerlifters [54,58]. Moreover, differences between populations were found in load–velocity profiles (e.g., young men vs. middle-aged men, or young men vs. young women) [30,59,60]. In addition, there are no studies comparing men and women in regard to load–velocity profiles in exercises such as hip-thrust, deadlifts or bench prone row. We encourage researchers to conduct further studies to determine whether these differences appear in other exercises. Furthermore, we consider it interesting to compare subjects with similar relative strength levels and athletes from different sports (e.g., football players vs. basketball players).

Although the results of this meta-analysis should be interpreted with caution due to the limitations mentioned above, the large number of studies that have been published on velocity-based training should be considered. This study has important practical applications for the prescription and monitoring of training loads in resistance training. Consequently, we consider that the results could have great potential for strength and conditioning coaches who can develop individualized training for women.

## 5. Conclusions

We conclude that there exist sex differences in mean propulsive velocity with light and moderate loads (30 and 70% 1RM) in bench press and squat exercises. Although these differences disappear for high loads (90% of 1RM), the average of male performance with all loads continues to show higher mean propulsive velocities. In the same line, independently of the muscle group, men display higher velocities, especially for upper limb exercises.

What seems to be clear is that we need to use specific equations based on force–velocity profiles for women. Finally, it would be interesting to conduct more studies on women regarding velocity-based training and strength so that conditioning coaches could prescribe training intensity more accurately and individualize training for the female population.

## Figures and Tables

**Figure 1 sports-11-00118-f001:**
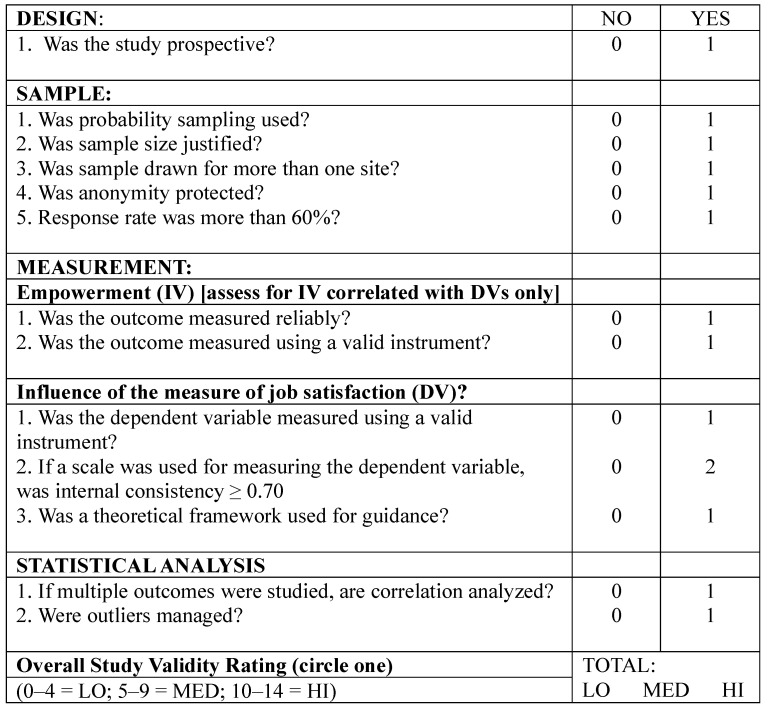
Quality assessment and validity tool for correlational studies.

**Figure 2 sports-11-00118-f002:**
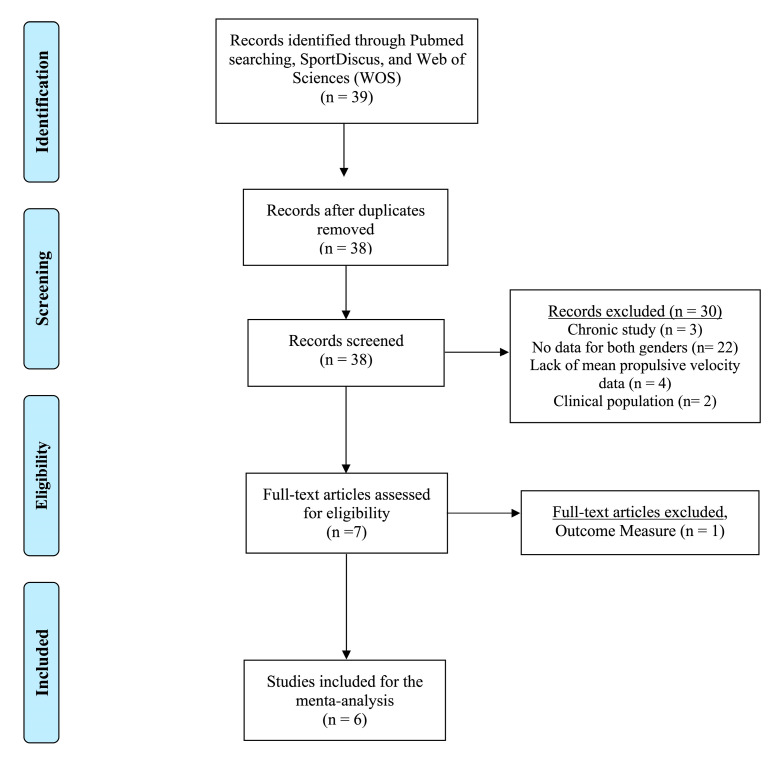
Flow diagram of the search process.

**Figure 3 sports-11-00118-f003:**
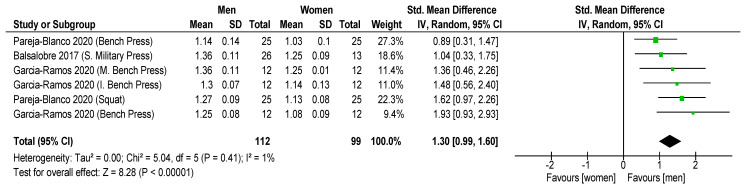
Comparison of Mean Propulsive Velocity in men vs. women for 30% 1RM. Forest plots show standardized mean differences with 95% confidence intervals (CI). The diamond at the bottom presents the pooled effect [29,35,31].

**Figure 4 sports-11-00118-f004:**
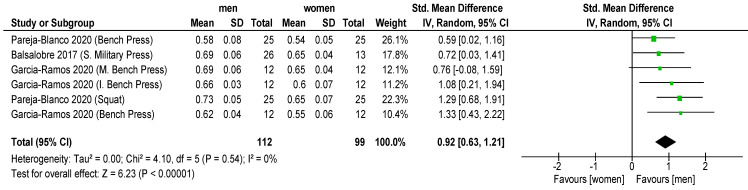
The forest plot from the meta-analysis of MVP in women vs. MVP in men for 70% 1RM. The *x*-axis denotes Cohen’s d (ES) while the whiskers denote the 95% CI. CI = confidence interval; MPV = mean propulsive velocity [29,35,31].

**Figure 5 sports-11-00118-f005:**
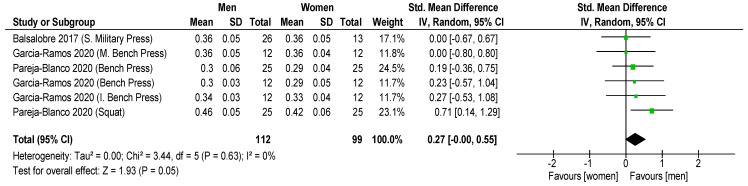
The forest plot from the meta-analysis of MVP in women vs. MVP in men for 90% 1RM. The *x*-axis denotes Cohen’s d (ES) while the whiskers denote the 95% CI. CI = confidence interval; MPV = mean propulsive velocity [35,31,29].

**Figure 6 sports-11-00118-f006:**
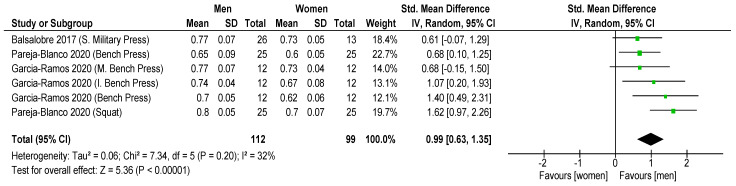
The forest plot from the meta-analysis of MVP in women vs. MVP in men for mean velocities. The *x*-axis denotes Cohen’s d (ES) while the whiskers denote the 95% CI. CI = confidence interval; MPV = mean propulsive velocity [29,31,35].

**Table 2 sports-11-00118-t002:** Results of quality assessment.

Reference	Item 1	Item 2	Item 3	Item 4	Item 5	Item 6	Item 7	Item 8	Item 9	Item 10	Item 11	Item 12	Item 13	Total Score
Pareja-Blanco et al. [29]	1	0	1	0	1	1	1	1	1	0	1	1	0	9
Torrejón et al. [30]	1	0	1	0	1	1	1	1	1	0	1	1	0	9
García-Ramos et al. [31]	1	1	1	0	1	1	1	1	1	0	1	1	1	11
Balsalobre et al. [35]	1	1	0	1	0	1	1	1	1	0	1	1	0	9
Iglesias-soler et al. [37]	1	1	0	1	0	1	1	1	1	1	1	1	0	10
Alonso-Aubin Diego A. et al. [36]	1	1	0	1	0	1	1	1	1	0	1	1	0	9

1 = criterion is satisfied; 0 = criterion is not satisfied. 0–4 low quality; 5–9 medium quality; 10–14 high quality.

## Data Availability

Not applicable.

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
