# Peer review of "A Systematic Review and Meta-Analysis of the Differences in Mean Propulsive Velocity between Men and Women in Different Exercises"

_sports, 2023, doi:10.3390/sports11060118_

Round 1
Reviewer 1 Report
This study sought to conduct a systematic review and meta-analysis of studies 16 examining the differences in the mean propulsive velocities between men and women in the different exercises studied (squat, bench press, inclined bench press and military press). This is a very great study. However, I have few comments for it.
-Abstract: please, to add more data (numbers) in results section.
-Introduction: Is not clear the hypothesis of Meta-Analysis. What was the hypothesis?
-Methods: The review followed the prisma (it is adequate). However, this MA was registred in prospero or another agency?
-Results: Bias was quantified? The authors could discuss it better.
-Conclusion: Using only 6 studies in MA and systematic review, is very difficult the authors conclude: "Finally, if we had specific velocities for women associated to each percentage of 1RM, strength and conditioning coaches could prescribe training intensity more accurately and individualize training for female athletes." Please, review it for clarification.
Reviewer 2 Report
Dear authors,
Thank you for the opportunity of reviewing this paper. I recognize the interest in this topic, and I would consider it for publication after some revisions.
Introduction
Line 42: I do not understand why Table 2 is mentioned. Please explain.
Lines 57-61: The authors mentioned that this study aims to verify the existence of significant differences in the mean propulsive velocities between men and women in several exercises. However, the previous background provided in the Introduction questions the estimation methods and validity of VBT in female athletes. Overall, the last paragraph of the Introduction should be rewritten. I also recommend including more details concerning the differences between male and female athletes in strength training. Finally, after reading the Introduction, it is unclear whether the authors are focused on athletes or the overall population.
Material & Methods
Line 66: Was the search protocol used previously recorded in any database, such as PROSPERO?
Line 71: It is not clear how the study selection was performed. How many authors participated in the search process? How many authors revised the papers selected?
Line 74: The search considered any date range?
Table 1: There are some abbreviations that are not mentioned in the footnotes. Please check. Also, in the training status, some studies present the sample characterization (judokas, rugby players) and others do not. Please include these details for each study.
Table 2: Where is the title?
Line 11: What does MPV mean? This is not explained before in the text.
Discussion
Line 77-79: “men have lower muscle mass”. Please confirm the sentence.
Limitations: The number of studies considered in this review represents a considerable limitation. Please clearly include this information in the limitation section. The number of studies considered for analysis should also be included in the abstract.
Overall, the Discussion is well-written and provides a good interpretation of the results. Congratulations.
